# Forensic Application of Monoclonal Anti-Human Glycophorin A Antibody in Samples from Decomposed Bodies to Establish Vitality of the Injuries. A Preliminary Experimental Study

**DOI:** 10.3390/healthcare9050514

**Published:** 2021-04-29

**Authors:** Benedetta Baldari, Simona Vittorio, Francesco Sessa, Luigi Cipolloni, Giuseppe Bertozzi, Margherita Neri, Santina Cantatore, Vittorio Fineschi, Mariarosaria Aromatario

**Affiliations:** 1Department of Anatomical, Histological, Forensic and Orthopedic Sciences, Sapienza University of Rome, 00161 Rome, Italy; benedetta.baldari@uniroma1.it (B.B.); simonavittorio1@gmail.com (S.V.); vittorio.fineschi@uniroma1.it (V.F.); 2Section of Legal Medicine, Department of Clinical and Experimental Medicine, University of Foggia, 71122 Foggia, Italy; francesco.sessa@unifg.it (F.S.); giuseppe.bertozzi@unifg.it (G.B.); santina.cantatore@unifg.it (S.C.); 3Department of Medical Sciences, University of Ferrara, 44121 Ferrara, Italy; margherita.neri@unife.it; 4Legal Medicine Division, Sant’Andrea University Hospital, 00189 Rome, Italy; maromatario@ospedalesantandrea.it

**Keywords:** forensic pathology, glycophorin A investigation, vitality

## Abstract

Glycophorins are an important group of red blood cell (RBC) transmembrane proteins. Monoclonal antibodies against GPA are employed in immunohistochemical staining during post-mortem examination: Through this method, it is possible to point out the RBC presence in tissues. This experimental study aims to investigate anti-GPA immunohistochemical staining in order to evaluate the vitality of the lesion from corpses in different decomposition state. Six cases were selected, analyzing autopsies’ documentation performed by the Institute of Legal Medicine of Rome in 2010–2018: four samples of fractured bones and three samples of soft tissues. For the control case, the fracture region of the femur was sampled. The results of the present study confirm the preliminary results of other studies, remarking the importance of the GPA immunohistochemical staining to highlight signs of survival. Moreover, this study suggests that the use of this technique should be routinely applied in cases of corpses with advanced putrefaction phenomena, even when the radiological investigation is performed, the macroscopic investigation is inconclusive, the H&E staining is not reliable. This experimental application demonstrated that the use of monoclonal antibody anti-human GPA on bone fractures and soft tissues could be important to verify whether the lesion is vital or not.

## 1. Introduction

Glycophorin (GP) A, GPB, GPC, and GPD are an important group of red blood cell (RBC) transmembrane proteins, described for the first time by Fairbanks et al. [1]. The four varieties of glycophorin represent approximately 2% of the total red blood cells (RBC) membrane protein: among the others, the predominant RBC glycophorin is GPA (glycophorin A—0.5 million copies per RBC). These proteins are characterized by a high content of sialic acid, giving a negative surface charge to the RBC [2]. For this reason, glycophorins play a pivotal role in controlling several essential functions such as the interactions both with vascular endothelium and with other blood cells. Moreover, GPA and B characterize the antigenic determinants, respectively, for the MN and Ss blood groups. Although the molecular function of GPA remains not fully understood, several studies demonstrate that it interacts with band 3, the significant erythrocytes membrane-spanning protein [3].

On the other hand, as red cell antigens, GPA can act as a binding site for specific antibodies. Monoclonal antibodies against glycophorin A (GPA), indeed, are frequently used in immunophenotypic studies for the identification of erythroid precursors in hematologic disorders. In the forensic field, the anti-human glycophorin A (GPA) monoclonal antibodies are commonly used in human blood detection, identifying red blood cell membrane antigen [4]. Moreover, monoclonal antibodies anti-GPA can be used in immunohistochemical staining at post-mortem examination to point out RBC presence in tissues [5]. Thus, if in non-putrefied corpses, the macroscopic evidence of hemorrhagic tissue infiltration is commonly considered a macroscopic sign of the viability of a lesion, research using anti-GPA antibodies could allow the differential diagnosis between ante-mortem and post-mortem lesions. This differentiation, indeed, is just as important as the identification of the exact cause of death, remaining the most notable challenge for the forensic pathologist. In this scenario, the definition of the vitality of skin and bone injuries is inscribed. For example, from a macroscopic perspective, it is well-known that the red-purplish coloration of a cut or bruise in fresh skin is indicative of its vitality. In more difficult cases, the microscopic investigation can be performed even just to confirm the macroscopic data [6]. Contrariwise, in the case of decomposed corpses, histological and immunohistochemical investigation can be considered mandatory: the transformative phenomena could mask or conceal the signs of traumatic injuries [7,8,9,10]. As previously described, a large number of immunohistochemical staining is frequently used in order to establish the vitality in skin lesions, such as anti-α-1 chymotrypsin, anti-fibronectin, anti-TGF-α and TGF-ß1, anti-inflammatory cytokines, anti-TNFα, anti-adhesion molecules, and anti-tryptase [11,12,13,14,15,16,17,18,19]. However, these antibodies show reactivity on well-preserved tissues, while their use remains uncertain when the post -mortem examination is performed in decomposed corpses or in human skeletal remains. Nevertheless, in the forensic field, the application on human bones of the immunohistochemical staining through the GPA antibody is not completely investigated.

In this setting, this experimental study aims to investigate anti-GPA antibody immunohistochemical staining in order to evaluate the vitality both of soft tissue injuries and bone fractures sampled from corpses in advanced decomposition state. Moreover, a comparison, between the hematoxylin-eosin (H&E) staining and anti-human GPA antibody immunohistochemistry technique, is also provided in the present manuscript to evaluate their potential in identifying “vital” processes in putrefied soft-tissue-free bone and in soft tissues.

## 2. Materials and Methods

All cases were collected among the documentation of all autopsies performed by the Institute of Legal Medicine of Rome from 2010 to 2018 (about 1500 autopsies). Six cases were selected on the following criteria: unnatural death, traumatic blunt injuries, body conservation status (almost chromatic phase of putrefaction and no macrofauna-related lesions), estimated post-mortem interval. Three cases were selected as control cases, dealing with a subject who died from drowning with a femur fracture occurred in the post-mortem period, during the translation operations, and two experimentally produced injuries (skin and soft tissues surrounding laryngeal area). The main characteristics of each case were summarized in Table 1.

For the purposes of this experimental study, the following tissues were sampled.

Case 1 dealt with a 57-year-old woman who died after a failed tentative of sexual assault. Several fractures were found affecting the cranial region. The external marginal area of the occipital fracture was sampled.

Case 2 concerned a 30-year-old male who died from other than traumatic causes. During the autopsy, a complete fracture of vertebral D4 was detected, whose sample was taken.

Case 3 was about a 39-year-old woman who died from acute cardio-respiratory insufficiency due to induced asphyxiation. Different injuries were found on the head. For the present study, a mandible fragment was sampled in the correspondence of the fracture.

Case 4 described the case of a 46-year-old woman who died from acute cardio-respiratory insufficiency from strangulation. Soft tissue surrounding laryngeal fracture was sampled.

Case 5 regarded a 37-year-old woman who died for homicidal violent mechanical asphyxiation; for this study, two tissues were collected: one sample from the II right rib, which was found fractured, and the other one from the neck skin, where signs of external compression were found.

Case 6 described the case of a three-month-old female, who died from cardiovascular arrest subsequent to massive brain edema, following a head injury. Considered the presence of bilateral retinal hemorrhages, the right retina tissue was sampled for the purposes of this study.

Finally, three cases were enrolled as negative controls: the case of a nine-year-old female, who died from drowning (Control 1). The corpse was exhumed six months after the burial. Before the autopsy a PMCT was performed, resulting negative for any injury. However, a femur-displaced fracture occurred in the post-mortem period, during the translation operations to the autopsy room. The lesions used as negative control of neck skin (Control 2) and tracheal soft tissues (Control 3) were experimentally produced.

At the end of the case selection, the following tissues were analyzed: four samples of fractured bones (cranium, vertebra, mandible, and rib) and three samples of different soft tissues (soft tissue of larynx, neck skin, and retina). Moreover, the fractured region of the femur (Control 1) was used as negative control case of bone injuries, while neck skin and tracheal soft tissues were used as negative control of soft tissue lesions.

Score rank, used in the present study to evaluate the force of immunohistochemical staining with anti-body anti-GPA, lied in a range from “negative” (marked as “−”) to “positive” (indicated with “+”).

In all cases, local prosecutors ordered autopsies to be performed to clarify the exact cause of death.

For all selected cases, H&E staining and anti-human GPA antibody immunohistochemistry investigation were performed both to establish RBC presence as vitality sign. In the present study, these stainings were performed on bones and on soft tissues for cases number 1, 2, 3, and 5. In case 4, the soft tissue of larynx was tested, while in case 6, the staining was performed in the retina.

On all sample collected during autopsy, 4 µm-thick paraffin-embedded sections were cut and stained with H&E staining following the standard protocol [20]. Staining with H&E was qualitatively classified as “reliable” (++), “not reliable” (--), and “not univocal” (+-), based on RBC morphologic identifiability.

In addition, anti-human GPA antibody immunohistochemistry investigation was performed using antibodies anti-glycophorin A (Santa Cruz Biotechnology, Inc., Dallas, TX, USA). The paraffin sections were mounted on slides covered with 3-aminopropyltriethoxysilane (Fluka, Buchs, Switzerland). Pre-treatment was necessary to facilitate antigen retrieval and to increase membrane permeability to antibodies anti-glycophorin A boiling 0.25 M EDTA buffer, at 20 °C. The primary antibody was applied in 1:500 ratio for glycophorin A and incubated for 120 min at 20 °C. The detection system utilized was the LSAB+ kit (Dako, Copenhagen, Denmark), a refined avidin-biotin technique in which a biotinylated secondary antibody reacts with several peroxidase-conjugated streptavidin molecules. The sections were counterstained with Mayer’s haematoxylin, dehydrated, cover slipped, and observed in a Leica DM4000B optical microscope (Leica, Cambridge, UK).

## 3. Results

In the present study, six cases were selected: five women and one male. In four cases, the corpses were found in advanced decomposition state (cases numbered 1, 2, 4, and 5). The histological investigations were very complex, considering the post-mortem transformative processes. Contrariwise, the corpses were in early stage of putrefaction, characterized by discoloration of the abdomen up to marbling phenomena in cases 3 and 6.

### 3.1. Macroscopic Analysis

The macroscopic analysis did not allow ascertaining the exact cause of death; for this reason, the microscopic investigation was performed. Concerning the cause of death, in three cases, the asphyxiation was defined as the manner of death (cases numbered 3, 4, and 5); in two cases (cases numbered 1 and 6), the cranial trauma was identified as the decisive damage. Finally, in case numbered 2, the victim died from acute cardio-respiratory distress; the death was not related to direct trauma.

### 3.2. Hemorrhagic Infiltration

Hemorrhagic infiltration was not identified at macroscopical analysis in the cases characterized by the transformative phenomena. Consequently, it was necessary to analyze and collect soft tissues and bone samples, performing the microscopic investigation. The results are summarized in Table 2.

H&E investigations were performed in order to highlight the RBC presence microscopically (Figure 1). In cases, numbers 1, 2, 4, and 5, hemorrhagic infiltrations were not found. Results were influenced by the postmortem transformative processes of tissues and bones.

In cases 3 and 6, hemorrhagic infiltrations were macroscopically detected near the mandible fracture (case 3) and on the retinal tissue (case 6); moreover, these findings were also confirmed with the H&E analysis, highlighting the presence of RBC.

### 3.3. Anti-Human GPA

The anti-human GPA antibody immunohistochemistry investigation was performed in all cases in order to establish the RBC presence and to evaluate the sign of vitality in the detected lesions (Figure 2). Even if this technique confirmed the presence of the RBC in all sampled tissues, it was considered fundamental to determine the exact cause of death in four of the six cases selected.

In case 1, the corpse was found in an advanced stage of decomposition and putrefaction. No radiological evidence of fracture was diagnosed; moreover, no macroscopical hemorrhagic infiltration was observed at pericranial soft tissue examination. However, a cranial linear compound fracture was found during the autopsy. H&E staining was not significant. Therefore, no evidence allows establishing if the fracture was caused by ante-mortem or post-mortem injury.

The GPA immunohistochemical investigation highlighted the presence of erythrocytes in the context of tissue cytoarchitecture, suggesting that the cranial fracture was vital, in accordance with the suspect’s declaration.

In cases 2 and 4, the corpses were found with advanced putrefactive phenomena. In both cases, no PMCT was performed. A vertebral fracture was found in case 2; in case 4, a suspicious area near the soft tissue of larynx was sampled. No hemorrhagic infiltration was noticed macroscopically in the two corpses, no unique interpretation was concluded after H&E staining examination. For these reasons, the immunohistochemistry was performed, confirming the presence of RBC, as the vitality of the lesion.

In case 5, the post-mortem decomposition was detected. PMCT and H&E investigations were performed, even if no important evidence was collected in order to establish the exact cause of death. The anti-GPA antibody research on the neck skin sampled was performed in order to obtain data about the lesion, resulting positive. The presence of erythrocytes allows confirming that the injury occurred when the victim was alive.

Contrariwise, in cases 3 and 6, the corpses presented only early decomposition stages. No PMTC was performed. Hemorrhagic infiltration was detected both at macroscopical and microscopical (H&E staining) investigation. The immunohistochemical anti-GPA analysis was positive, supporting other evidence about the ante-mortem origin of the injury.

Finally, the control cases were performed on different tissues (Figure 3), produced during the translation operations to the autopsy room. No reaction to the GPA immunohistochemical analysis was described.

## 4. Discussion

Glycophorin is an integral membrane protein on the plasma membrane of erythrocytes. Monoclonal antibodies against glycophorin A are frequently used in clinical medicine for the identification of erythroid precursors in hematologic disorders [21]; moreover, this immunohistochemical staining can be used in forensic pathology for the identification of RBC in bone and tissue. According to the literature, to date, forensic application of the GPA immunohistochemical technique in order to evaluate the vitality of a wound is not quite investigated. As described by Cattaneo et al., the presence of clots and red blood cell residues on the fractured margins can be considered strongly indicative of vital reaction [6].

However, in this study, anti-glycophorin A analysis was performed on bones (cases number 1, 2 3, and 5) and on soft tissues sampled in corpses found in different postmortem period: This technique was applied on the soft tissue of larynx in case 4, on wrinkle neck in case 5, and on the retina in case 6.

Technical difficulties were described in handling decomposed bodies, considering the presence of artifactual alteration of tissue structure and microscopic features [22]. In the discussed study, hemorrhagic infiltration is not macroscopically evident in decomposed corpses (cases 1, 2, 4, and 5). Nevertheless, at the macroscopical examination, it was possible to observe the presence of hemorrhagic infiltration on the mandible fracture (case 3) and in the retinal tissue (case 6). It is essential to note that the bodies sampled in cases 3 and 6 were well preserved: In these cases, it was simple to detect hemorrhagic infiltration during autopsies.

Although it is likely that more information may be gleaned from fresh bodies in perfectly preserved states, decomposed bodies may reveal significant anatomical and pathological features that enable both the cause and manner of death to be established.

Even if it is well described that histopathology techniques can be used in the identification of the vitality of lesions in preserved tissue, the efficacy of these techniques in putrefied tissue is not demonstrated. In the present study, H&E investigation was not reliable in cases numbered 1, 2, 4, and 5: Considering that the corpses were recovered in the 5th stage of human decomposition process, it was impossible to define the presence of hemorrhagic infiltration. On the other hand, this technique was able to demonstrate the presence of RBC in the tissues sampled from cases number 3 and 6 (early stage of human decomposition process).

The immunohistochemical investigation was performed in all cases, showing interesting results. Monoclonal antibodies against GPA resulted positive in all analyzed cases, indicating the presence of RBC and demonstrating the vitality at the moment of the lesion. Indeed, the results in the control case were negative.

The discussed data confirmed that the histopathological investigation should be combined with the immunohistochemical examination: Indeed, evaluating the vitality of an injury, immunohistochemical diagnosis can provide reliable information [23]. Mainly, this study highlighted the importance of the GPA technique both on bones and on soft tissue in order to collect information on RBC presence, collecting information about the vitality of the lesion.

Bone fracture is described as a complete or incomplete disruption of bone tissue continuity: When it occurs during the life, the fracture triggers a regular tissue reaction [24,25,26]. Although fracture healing depends on the age of the individual and their nutritional status, age does not play an important role once adulthood has been reached [27]. The histological finding can demonstrate particular features in fractures following blast trauma [28]. Macroscopic morphological patterns of bone fracture are routinely used in forensic pathology and anthropology to distinguish between ante-mortem, peri-mortem, and postmortem injuries. Based on macroscopic and microscopic findings, it is possible to classify the fracture, avoiding inaccuracy [29]. Indeed, even if the presence of erythrocytes does not prove the vitality of the reaction, it could be used as a marker of bleeding to suggest another confirmative investigation on the vitality of injuries.

According to this experimental study, the glycophorin analysis is very important in corpses found in advanced decomposition state. Particularly, it could be very helpful when there is no evidence of hemorrhagic infiltrations, both at macroscopical analysis and histological investigation.

Even if in forensic pathology the use of the anti-GPA immunohistochemical staining remains controversial, according to the presented results, this immunohistochemical technique can be considered very useful both on bone structures and on soft tissues, particularly when the corpses have been found in an advanced state of decomposition. A negative result suggests the absence of erythrocytes in the sampled tissue: This means that the lesion could be related to post mortem trauma. On the contrary, a positive result can suggest the presence of erythrocytes in the tissue, ascertain that the lesion on bone and/or on soft tissue has been generated by pre or perimortem trauma. In a recent study, it was described the usability for forensic purposes of the immunochemical staining in corpses with different conservation status [30], as confirmed by this study. In addition, the presented results are in line with Taborelli at al.’s [5] research article on glycophorin A, as a good marker to orient diagnosis of vitality thanks to the high resistance at different PMI. Moreover, this study showed that this immunohistochemical investigation is able to keep its diagnostic reliability in different types of tissue. Therefore, this methodological approach should be routinely applied in corpses with advanced putrefaction phenomena, even when the radiological investigation is performed, the macroscopic investigation is inconclusive, and the H&E staining is not reliable. Agreeing with observation of Cappella et al. [31], this study confirms that searching for bleeding or tissue reactions in fragmented bones or in tissues with advanced taphonomy is useful to reach a diagnosis of ante-mortem or post-mortem injury. Considering the statement of the same authors, we argue that simultaneous application of histological and immuno-histochemical methods searching for inflammatory markers and hemorrhaging joined with research of bone reaction activity is useful to reach a sustainable diagnosis of vitality.

## 5. Conclusions

In conclusion, though certainly not conclusive, this experimental application demonstrated that the use of monoclonal antibody anti-human GPA on bone fractures as well as soft tissues could be important to verify whether the lesion is vital or not.

## Figures and Tables

**Figure 1 healthcare-09-00514-f001:**
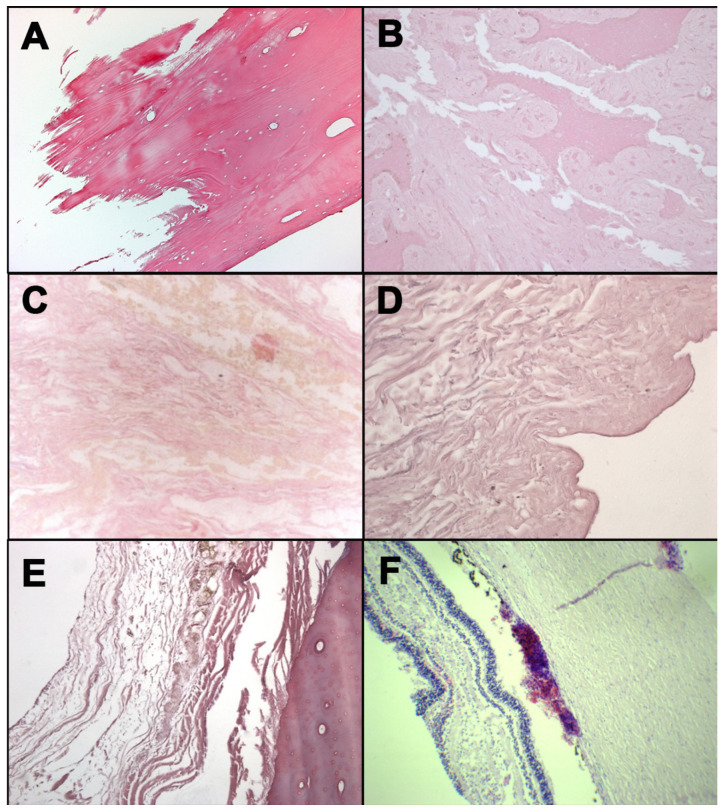
H&E staining: occipital bone from case 1 (**A**); mandible fragment from case 3 (**B**); soft tissues surrounding laryngeal fracture from case 4 (**C**); neck skin from case 5 (**D**); rib fracture from case 5 (**E**); retina from case 6 (**F**).

**Figure 2 healthcare-09-00514-f002:**
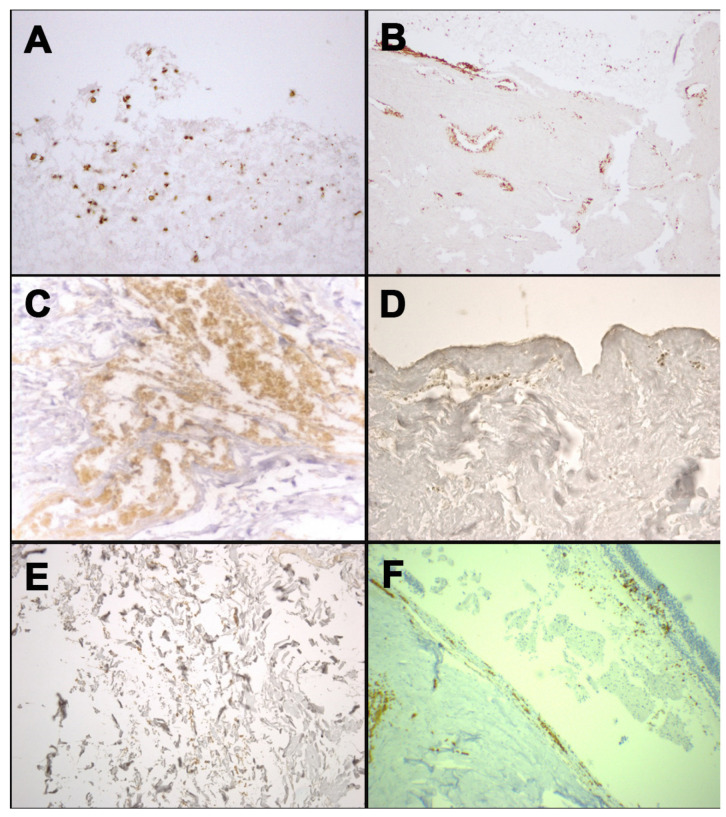
Immunohistochemical staining with anti-human GPA antibody: positivity at the margin of occipital bone fracture from case 1 (**A**); positivity at the stump of the mandible fracture from case 3 (**B**); positivity in the context of soft tissues surrounding laryngeal fracture from case 4 (**C**); positivity in dermis of neck skin from case 5 (**D**); positivity in the contest of soft tissues surrounding rib fracture from case 5 (**E**); positivity of the retinal sample from case 6 (**F**).

**Figure 3 healthcare-09-00514-f003:**
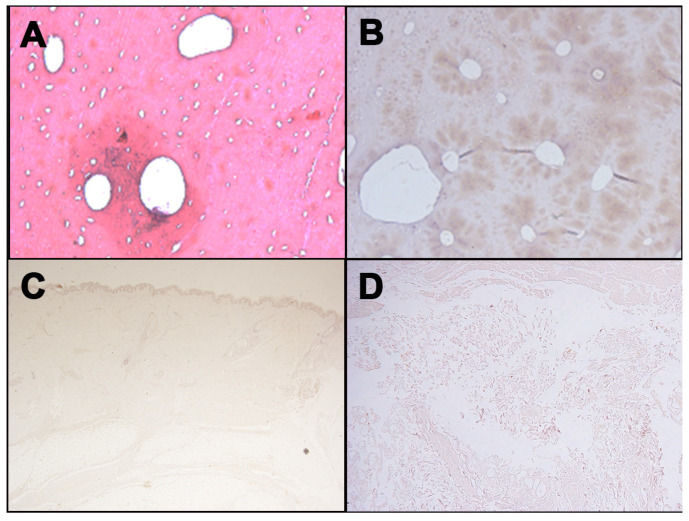
Negative immunohistochemical staining with anti-human GPA antibody for different tissues: Femur ((**A**), H&E; (**B**), immunostaining with anti-human GPA antibody); (**C**) Skin; (**D**) Soft tissues surrounding laryngeal area.

**Table 1 healthcare-09-00514-t001:** The main characteristics of the selected case. PMCT: Post-mortem computed tomography; PMI: Post-mortem interval; CTL: Control.

Case (Tissue)	Sex and Age	Type of Offence	PMCT	State	Estimated PMI
**1** **(cranial)**	F 57 year	Homicide	Yes	Later stage of Putrefaction	65 days
**2** **(vertebral)**	M 30 year	Accident	No	Later stage of Putrefaction	70 days
**3** **(mandible)**	F 39 year	Homicide	No	Early stage of Putrefaction	2 days
**4** **(larynx)**	F 46 year	Homicide	No	Later stage of Putrefaction	181 days
**5** **(neck and rib)**	F 37 year	Homicide	Yes	Later stage of Putrefaction	85 days
**6** **(retina)**	F 3 months	Homicide	No	Early stage of Putrefaction	7 days
**CTL 1**	F 9 year	Accidental	Yes	Later stage of Putrefaction	187 days
**CTL 2**	M 29 year	Experimental	Yes	Early stage of Putrefaction	15 days
**CTL 3**	M 52 year	Experimental	No	Later stage of Putrefaction	45 days

**Table 2 healthcare-09-00514-t002:** The main results of the histological and immunoistochemical examination.

Case	H&E	Glycophorin
**1 (cranial)**	No Reliable	Positive on cranial fracture
**2 (vertebral)**	No Reliable	Positive on vertebral fracture
**3 (mandible)**	Reliable	Positive on mandible fracture
**4 (larynx)**	Not Univocal	Positive on soft tissue of larynx
**5 (neck and rib)**	No Reliable	Positive on wrinkle neck and rib
**6 (retina)**	Reliable	Positive on retina
**CTL (different tissues)**	Reliable	Negative on post-mortem fracture

## Data Availability

All data are included in the manuscript.

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
