# Peer review of "Forensic Application of Monoclonal Anti-Human Glycophorin A Antibody in Samples from Decomposed Bodies to Establish Vitality of the Injuries. A Preliminary Experimental Study"

_healthcare, 2021, doi:10.3390/healthcare9050514_

Round 1

Reviewer 1 Report

The article presents a study on 6 real cases of determination of the viability of bone and soft tissue lesions through the use of monoclonal antibodies to glycophorin A. The perspective provided by the article is certainly of interest in the field of forensic science and in particular in forensic pathology. I believe that the article is worthy of publication, even if I suggest to the authors some modifications and some clarifications.

I would ask the Authors to clarify how the define H&F results as “No reliable”, “Reliable” and “Not univocal”. It should be clarified in the Materials and Methods section.

I have some doubts about how the control was chosen. How was it ascertained that the fracture occurred during transport to the autopsy room? Were radiological investigations done before and after autopsy? I think it is useful to specify how the control was chosen.

Here some minor issues:

  • Line 59: “betweenante-mortem” should be changed in “between ante-mortem”-
  • Lines 126-133: I think these sentences are not part of the manuscript, maybe it’s something belonging to the Instructions for Authors?
  • Lines 137-138: I think that in this sentence the verb is missing, maybe the correct sentence is “ H&E staining and anti-human GPA antibody immunohistochemistry investigation were performed both to establish …”
  • Line 321: Institutional Review Board Statement, Authors state that this is not applicable. What about the approval you cite at lines 124-125 (Scientific Committee of University of Roma)?

Author Response

Dear Reviewer,

thank for spending your time to revise our paper and helping us to make it better.

in order to your reliefs:

1) we clarify our criteria of definition of H&E results;

2) in order to time of production of fracture in control 1 we performed a PMTC after whom the corpse accidentally fallen producing the fractures that was not detected by TC; we addedd furthe control-cases, as suggested by others reviewers clarifying how they have been choosen

3) Line 59: “betweenante-mortem”: it has been correct;

4) Lines 126-133: indeed the phrase was not part of the manuscript, sorry.

5)  Lines 137-138: "... were performed .." has been added

6) Line 321: Institutional Review Board Statement: indeed the phrase was not part of the manuscript and it was deleted, sorry

Reviewer 2 Report

The authors present a research article in which they demonstrate, through the experimental application of the GPA immunohistochemical staining on 4 samples of fractured bones and 3 samples of soft tissues, that the use of monoclonal antibody anti-human GPA could be important to verify whether the lesion is vital or not on bone fractures and soft tissues.

The idea behind the study is certainly o interest for the forensic community although not quite original. There is a very thorough review wrote by Cappella A. and Cattaneo C., entitled “Exiting the limbo of perimortem trauma: A brief review of microscopic markers of hemorrhaging and early healing signs in bone” published in FSI in 2019 that comments of the strengths and limits of histology and immunohistochemistry for diagnosing vitality.  

Major points

1) the number of cases included (only 4) is too limited.

2) the number of controls (only one) in inappropriate. The authors should at least pair each case to a control of the same tissue/organ (i.e. skin versus skin; bone vs bone; retina vs retina)

3) The inclusion criteria (page 2) should be better clarified;

4) A semi-quantitative method should be used to evaluate Glycophorin positivity

5) The results should be clearer presented and discussed in comparison to previous literature findings (see Cappella et al. for example)

6) An extensive language editing is necessary

Author Response

Dear Reviwer,

thank for spending your time to review our paper and helpingo to do it better.

In order to your suggestions:

1) The number of cases included (only 4) is too limited: the cases reported, really, are six; in any case this is an experimental pilot study and no similar casuistic are reported in literature. We aim to expand the number of cases in future papers.

2) "... the number of controls (only one) in inappropriate. The authors should at least pair each case to a control of the same tissue/organ (i.e. skin versus skin; bone vs bone; retina vs retina) ...". Thank you for this suggestion: we addedd further controls about skin and soft tissues taken from other cases we observed in our Institute.

3) " ...The inclusion criteria (page 2) should be better clarified ...": inclusion criteria have been integrated and better specified;

4) "... A semi-quantitative method should be used to evaluate Glycophorin positivity ...": the method of evaluation of Glycophorin with semiquantitative score has been added to material and methods (Score rank, used in the present study to evaluate the force of immunohistochemical staining with anti-body anti-GPA,  lied in a range from “negative” (marked as “−”) to “positive” (indicated with “+”). ...)

5) " The results should be clearer presented and discussed in comparison to previous literature findings (see Cappella et al. for example) ..": the discussion has been integrated even with reference to previous literature and a comaprison with it has been performed

 6) "... An extensive language editing is necessary ...": editing of english has been performed

Reviewer 3 Report

The subject of the article is interesting, once demonstrated that the use of monoclonal antibody anti-human GPA on bone fractures and soft tissues could be important biomarker in tissues in advanced putrefactive state. However, more controls are need to prove the preliminary data.

General comments:

  1. In the introduction, the firsts sentence “Glycophorin (GP) A, GPB, GPC, and GPD are an important group of red blood cell (RBC) transmembrane proteins, responsible for various physiological functions.” which functions, please list the main ones.

  1. The procedures were approved by the Scientific Committee of University of Roma (Ita- 124ly–RMHH_20/05/2019) or by the Ethics Committees of University of Roma?? all procedures with human, must have the ethics committee's approval.

  1. I think the date of death until the collection of the tissue is a very important element, to understand the advanced putrefactive.
  2. The results cannot be included in the introduction, please rephrase the last sentence of the introduction.
  3. In my opinion the biggest problem of the article, is that the control are only from femoral fracture. The authors should have more tissues from control. So, I think they could include more results from the control (retinal sample, soft tissues surrounding rib fracture, occipital bone….) with different putrefactive states.

Author Response

Dear Reviewer

We thank you for yourtime and helpful comments improvingscientific quality of our manuscript. Thanking you again for all your help and consideration, I remain with my best personal regard.

"... The subject of the article is interesting, once demonstrated that the use of monoclonal antibody anti-human GPA on bone fractures and soft tissues could be important biomarker in tissues in advanced putrefactive state. However, more controls are need to prove the preliminary data ...".

 Thank you for your positive comment.

General comments:

  1. In the introduction, the firsts sentence “Glycophorin (GP) A, GPB, GPC, and GPD are an important group of red blood cell (RBC) transmembrane proteins, responsible for various physiological functions.” which functions, please list the main ones.

Thank you for your suggestion.In order to avoid repetitions, it was preferred to modify the first sentence, as the GP functions were mentioned in the following lines.

  1. The procedures were approved by the Scientific Committee of University of Roma (Ita- 124ly–RMHH_20/05/2019) or by the Ethics Committees of University of Roma?? all procedures with human, must have the ethics committee's approval.

This investigation conforms to the principles outlined in the declaration of Helsinki.            

  1. I think the date of death until the collection of the tissue is a very important element, to understand the advanced putrefactive.

Thank you for your comment.In Table 1 an estimation of time from death is provided. If it is not sufficient let us know how to improve the manuscript.

  1. The results cannot be included in the introduction, please rephrase the last sentence of the introduction.

Thank you for your suggestion. The last sentence has been rephrased according to your comment.

  1. In my opinion the biggest problem of the article, is that the control are only from femoral fracture. The authors should have more tissues from control. So, I think they could include more results from the control (retinal sample, soft tissues surrounding rib fracture, occipital bone….) with different putrefactive states.

Thank you for your suggestion. We have added the requested information in the text.

Please do not hesitate to contact me for any further questions.

Yours Sincerely.

Round 2

Reviewer 3 Report

The authors have revised the manuscript according to the reviewer's comments.

I recommend accept.

Author Response

Thank you for your suggestions that allow to get the paper better!